# ReAct: Out-of-distribution Detection With Rectified Activations

**Yiyou Sun**
Department of Computer Sciences
University of Wisconsin-Madison
sunyiyou@cs.wisc.edu

**Chuan Guo**
Facebook AI Research
chuanguo@fb.com

**Yixuan Li**
Department of Computer Sciences
University of Wisconsin-Madison
sharonli@cs.wisc.edu

## Abstract

Out-of-distribution (OOD) detection has received much attention lately due to its practical importance in enhancing the safe deployment of neural networks. One of the primary challenges is that models often produce highly confident predictions on OOD data, which undermines the driving principle in OOD detection that the model should only be confident about in-distribution samples. In this work, we propose **ReAct**—a simple and effective technique for reducing model overconfidence on OOD data. Our method is motivated by novel analysis on internal activations of neural networks, which displays highly distinctive signature patterns for OOD distributions. Our method can generalize effectively to different network architectures and different OOD detection scores. We empirically demonstrate that ReAct achieves competitive detection performance on a comprehensive suite of benchmark datasets, and give theoretical explication for our method's efficacy. On the ImageNet benchmark, ReAct reduces the false positive rate (FPR95) by 25.05% compared to the previous best method[1].

## 1   Introduction

Neural networks deployed in real-world systems often encounter out-of-distribution (OOD) inputs—unknown samples that the network has not been exposed to during training. Identifying and handling these OOD inputs can be paramount in safety-critical applications such as autonomous driving [9] and health care. For example, an autonomous vehicle may fail to recognize objects on the road that do not appear in its object detection model's training set, potentially leading to a crash. This can be prevented if the system identifies the unrecognized object as OOD and warns the driver in advance.

A driving idea behind OOD detection is that the model should be much more uncertain about samples outside of its training distribution. However, Nguyen et al. [38] revealed that modern neural networks can produce overconfident predictions on OOD inputs. This phenomenon renders the separation of in-distribution (ID) and OOD data a non-trivial task. Indeed, much of the prior work on OOD detection focused on defining more suitable measures of OOD uncertainty [17, 19, 28, 29, 31, 33]. Despite the improvement, it is arguable that continued research progress in OOD detection requires insights into the fundamental cause and mitigation of model overconfidence on OOD data.

In this paper, we start by revealing an important observation that OOD data can trigger unit activation patterns that are significantly different from ID data. Figure 1 (b) shows the distribution of activations

---

[1]Code is available at: https://github.com/deeplearning-wisc/react.git

35th Conference on Neural Information Processing Systems (NeurIPS 2021).

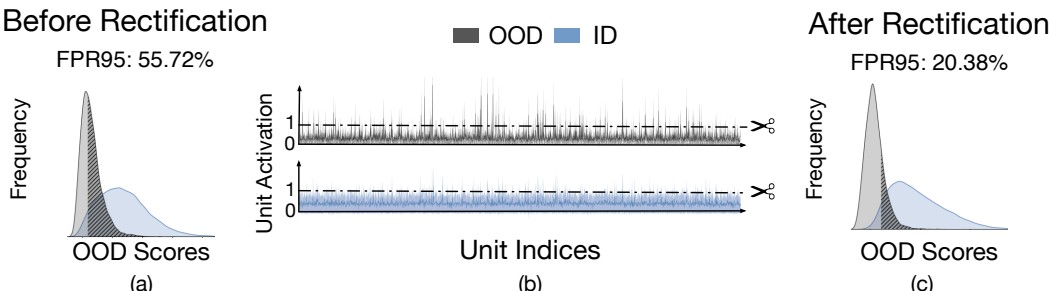

Figure 1: Plots showing (a) the distribution of ID (ImageNet [7]) and OOD (iNaturalist [16]) uncertainty scores before truncation, (b) the distribution of per-unit activations in the penultimate layer for ID and OOD data, and (c) the distribution of OOD uncertainty scores [33] after rectification. Applying ReAct drastically improves the separation of ID and OOD data. See text for details.

in the penultimate layer of ResNet-50 trained on ImageNet [7]. Each point on the horizontal axis corresponds to a single unit. The mean and standard deviation are shown by the solid line and shaded area, respectively. The mean activation for ID data (blue) is well-behaved with a near-constant mean and standard deviation. In contrast, for OOD data (gray), the mean activation has significantly larger variations across units and is biased towards having sharp positive values (*i.e.*, positively skewed). As a result, such high unit activation can undesirably manifest in model output, producing overconfident predictions on OOD data. A similar distributional property holds for other OOD datasets as well.

The above observation naturally inspires a simple yet surprisingly effective method—**Re**ctified **Act**ivations (dubbed **ReAct**) for OOD detection. In particular, the outsized activation of a few selected hidden units can be attenuated by rectifying the activations at an upper limit $c > 0$. Conveniently, this can be done on a pre-trained model without any modification to training. The dashed horizontal line in Figure 1(b) shows the cutoff point $c$, and its effect on the OOD uncertainty score is shown in Figure 1(c). After rectification, the output distributions for ID and OOD data become much more well-separated and the false positive rate (FPR) is significantly reduced from $55.72\%$ to $20.38\%$. Importantly, this truncation largely preserves the activation for in-distribution data, and therefore ensures the classification accuracy on the original task is largely comparable.

We provide both empirical and theoretical insights, characterizing and explaining the mechanism by which ReAct improves OOD detection. We perform extensive evaluations and establish state-of-the-art performance on a suite of common OOD detection benchmarks, including CIFAR-10 and CIFAR-100, as well as a large-scale ImageNet dataset [7]. ReAct outperforms the best baseline by a large margin, reducing the average FPR95 by up to $25.05\%$. We further analyze our method theoretically and show that ReAct is more beneficial when OOD activations are more chaotic (*i.e.*, having a larger variance) and positively skewed compared to ID activations, a behavior that is typical of many OOD datasets (*cf.* Figure 1). In summary, our **key results and contributions** are:

1. We introduce ReAct —a simple and effective *post hoc* OOD detection approach that utilizes activation truncation. We show that ReAct can generalize effectively to different network architectures and works with different OOD detection methods including MSP [15], ODIN [31], and energy score [33].

2. We extensively evaluate ReAct on a suite of OOD detection tasks and establish a state-of-the-art performance among post hoc methods. Compared to the previous best method, ReAct achieves an FPR95 reduction of $25.05\%$ on a large-scale ImageNet benchmark.

3. We provide both empirical ablation and theoretical analysis, revealing important insights that abnormally high activations on OOD data can harm their detection and how ReAct effectively mitigates this issue. We hope that our insights inspire future research to further examine the internal mechanisms of neural networks for OOD detection. Code and dataset will be released for reproducible research.

## 2 Background

The problem of *out-of-distribution detection* for classification is defined using the following setup. Let $\mathcal{X}$ (resp. $\mathcal{Y}$) be the input (resp. output) space and let $\mathcal{P}$ be a distribution over $\mathcal{X} \times \mathcal{Y}$. Let $f : \mathcal{X} \to \mathbb{R}^{|\mathcal{Y}|}$ be a neural network trained on samples drawn from $\mathcal{P}$ to output a logit vector, which is used to predict the label of an input sample. Denote by $\mathcal{D}_{\text{in}}$ the marginal distribution of $\mathcal{P}$ for $\mathcal{X}$, which represents the distribution of in-distribution data. At test time, the environment can present a distribution $\mathcal{D}_{\text{out}}$ over $\mathcal{X}$ of out-of-distribution data. The goal of OOD detection is to define a decision function $G$ such that for a given test input $\mathbf{x} \in \mathcal{X}$:

$$G(\mathbf{x}; f) = \begin{cases} 0 & \text{if } \mathbf{x} \sim \mathcal{D}_{\text{out}}, \\ 1 & \text{if } \mathbf{x} \sim \mathcal{D}_{\text{in}}. \end{cases}$$

The difficulty of OOD detection greatly depends on the separation between $\mathcal{D}_{\text{in}}$ and $\mathcal{D}_{\text{out}}$. In the extreme case, if $\mathcal{D}_{\text{in}} = \mathcal{D}_{\text{out}}$ then the optimal decision function is a random coin flip. In practice, $\mathcal{D}_{\text{out}}$ is often defined by a distribution that simulates unknowns encountered during deployment time, such as samples from an irrelevant distribution whose label set has no intersection with $\mathcal{Y}$ and therefore should not be predicted by the model.

## 3 Method

We introduce a simple and surprisingly effective technique, **Re**ctified **Act**ivations (ReAct), for improving OOD detection performance. Our key idea is to perform *post hoc* modification to the unit activation, so to bring the overall activation pattern closer to the well-behaved case. Specifically, we consider a pre-trained neural network parameterized by $\theta$, which encodes an input $\mathbf{x} \in \mathbb{R}^d$ to a feature space with dimension $m$. We denote by $h(\mathbf{x}) \in \mathbb{R}^m$ the feature vector from the penultimate layer of the network. A weight matrix $\mathbf{W} \in \mathbb{R}^{m \times K}$ connects the feature $h(\mathbf{x})$ to the output $f(\mathbf{x})$, where $K$ is the total number of classes in $\mathcal{Y} = \{1, 2, ..., K\}$.

**ReAct: Rectified Activation.** We propose the `ReAct` operation, which is applied on the penultimate layer of a network:

$$\bar{h}(\mathbf{x}) = \text{ReAct}(h(\mathbf{x}); c), \tag{1}$$

where $\text{ReAct}(x; c) = \min(x, c)$ and is applied element-wise to the feature vector $h(\mathbf{x})$. In effect, this operation truncates activations above $c$ to limit the effect of noise. The model output after *rectified activation* is given by:

$$f^{\text{ReAct}}(\mathbf{x}; \theta) = \mathbf{W}^\top \bar{h}(\mathbf{x}) + \mathbf{b}, \tag{2}$$

where $\mathbf{b} \in \mathbb{R}^K$ is the bias vector. A higher $c$ indicates a larger threshold of activation truncation. When $c = \infty$, the output becomes equivalent to the original output $f(\mathbf{x}; \theta)$ without rectification, where $f(\mathbf{x}; \theta) = \mathbf{W}^\top h(\mathbf{x}) + \mathbf{b}$. Ideally, the rectification parameter $c$ should be chosen to sufficiently preserve the activations for ID data while rectifying that of OOD data. In practice, we set $c$ based on the $p$-th percentile of activations estimated on the ID data. For example, when $p = 90$, it indicates that 90% percent of the ID activations are less than the threshold $c$. We discuss the effect of percentile in detail in Section 4.

**OOD detection with rectified activation.** During test time, ReAct can be leveraged by a variety of downstream OOD scoring functions relying on $f^{\text{ReAct}}(\mathbf{x}; \theta)$:

$$G_\lambda(\mathbf{x}; f^{\text{ReAct}}) = \begin{cases} \text{in} & S(\mathbf{x}; f^{\text{ReAct}}) \geq \lambda \\ \text{out} & S(\mathbf{x}; f^{\text{ReAct}}) < \lambda \end{cases}, \tag{3}$$

where a thresholding mechanism is exercised to distinguish between ID and OOD during test time. To align with the convention, samples with higher scores $S(\mathbf{x}; f)$ are classified as ID and vice versa. The threshold $\lambda$ is typically chosen so that a high fraction of ID data (*e.g.,* 95%) is correctly classified. ReAct can be compatible with several commonly used OOD scoring functions derived from the model output $f(\mathbf{x}; \theta)$, including the softmax confidence [15], ODIN score [31], and the energy score [33]. For readers' convenience, we provide an expansive description of these methods in Appendix B. In Section 4, we default to using the energy score (since it is hyperparameter-free and does not require fine-tuning), but demonstrate the benefit of using ReAct with other OOD scoring functions too.

| Model | Methods | OOD Datasets | | | | | | | | Average | |
|---|---|---|---|---|---|---|---|---|---|---|---|
| | | iNaturalist | | SUN | | Places | | Textures | | | |
| | | FPR95 ↓ | AUROC ↑ | FPR95 ↓ | AUROC ↑ | FPR95 ↓ | AUROC ↑ | FPR95 ↓ | AUROC ↑ | FPR95 ↓ | AUROC ↑ |
| ResNet | MSP [15] | 54.99 | 87.74 | 70.83 | 80.86 | 73.99 | 79.76 | 68.00 | 79.61 | 66.95 | 81.99 |
| | ODIN [31] | 47.66 | 89.66 | 60.15 | 84.59 | 67.89 | 81.78 | 50.23 | 85.62 | 56.48 | 85.41 |
| | Mahalanobis [29] | 97.00 | 52.65 | 98.50 | 42.41 | 98.40 | 41.79 | 55.80 | 85.01 | 87.43 | 55.47 |
| | Energy [33] | 55.72 | 89.95 | 59.26 | 85.89 | 64.92 | 82.86 | 53.72 | 85.99 | 58.41 | 86.17 |
| | **ReAct (Ours)** | **20.38** | **96.22** | **24.20** | **94.20** | **33.85** | **91.58** | **47.30** | **89.80** | **31.43** | **92.95** |
| MobileNet | MSP [15] | 64.29 | 85.32 | 77.02 | 77.10 | 79.23 | 76.27 | 73.51 | 77.30 | 73.51 | 79.00 |
| | ODIN [31] | 55.39 | 87.62 | 54.07 | 85.88 | 57.36 | 84.71 | 49.96 | 85.03 | 54.20 | 85.81 |
| | Mahalanobis [29] | 62.11 | 81.00 | 47.82 | 86.33 | 52.09 | 83.63 | 92.38 | 33.06 | 63.60 | 71.01 |
| | Energy [33] | 59.50 | 88.91 | 62.65 | 84.50 | 69.37 | 81.19 | 58.05 | 85.03 | 62.39 | 84.91 |
| | **ReAct (Ours)** | **42.40** | **91.53** | **47.69** | **88.16** | **51.56** | **86.64** | **38.42** | **91.53** | **45.02** | **89.47** |

Table 1: **Main results.** Comparison with competitive *post hoc* out-of-distribution detection methods. All methods are based on a model trained on **ID data only** (ImageNet-1k), without using any auxiliary outlier data. ↑ indicates larger values are better and ↓ indicates smaller values are better. All values are percentages.

## 4 Experiments

In this section, we evaluate ReAct on a suite of OOD detection tasks. We first evaluate a on large-scale OOD detection benchmark based on ImageNet [18] ( Section 4.1), and then proceed in Section 4.2 with CIFAR benchmarks [27].

### 4.1 Evaluation on Large-scale ImageNet Classification Networks

We first evaluate ReAct on a large-scale OOD detection benchmark developed in [18]. Compared to the CIFAR benchmarks that are routinely used in literature, the ImageNet benchmark is more challenging due to a larger label space ($K = 1,000$). Moreover, such large-scale evaluation is more relevant to real-world applications, where the deployed models often operate on images that have high resolution and contain more classes than the CIFAR benchmarks.

**Setup.** We use a pre-trained ResNet-50 model [12] for ImageNet-1k. At test time, all images are resized to $224 \times 224$. We evaluate on four test OOD datasets from (subsets of) Places365 [62], Textures [5], iNaturalist [16], and SUN [55] with non-overlapping categories w.r.t ImageNet. Our evaluations span a diverse range of domains including fine-grained images, scene images, and textural images (see Appendix C for details). We use a validation set of Gaussian noise images, which are generated by sampling from $\mathcal{N}(0, 1)$ for each pixel location. To ensure validity, we further verify the activation pattern under Gaussian noise, which exhibits a similar distributional trend with positive skewness and chaoticness; see Figure 7 in Appendix G for details. We select $p$ from $\{10, 65, 80, 85, 90, 95, 99\}$ based on the FPR95 performance. The optimal $p$ is 90. All experiments are based on the hardware described in Appendix D.

**Comparison with competitive OOD detection methods.** In Table 1, we compare ReAct with OOD detection methods that are competitive in the literature, where ReAct establishes the state-of-the-art performance. For a fair comparison, all methods use the pre-trained networks *post hoc*. We report performance for each OOD test dataset, as well as the average of the four. ReAct outperforms all baselines considered, including Maximum Softmax Probability [15], ODIN [31], Mahalanobis distance [29], and energy score [33]. Noticeably, ReAct reduces the FPR95 by **25.05**% compared to the best baseline [31] on ResNet. Note that Mahalanobis requires training a separate binary classifier, and displays limiting performance since the increased size of label space makes the class-conditional Gaussian density estimation less viable. In contrast, ReAct is much easier to use in practice, and can be implemented through a simple *post hoc* activation rectification.

**Effect of rectification threshold $c$.** We now characterize the effect of the rectification parameter $c$, which can be modulated by the percentile $p$ described in Section 3. In Table 2, we summarize the OOD detection performance, where we vary $p = \{10, 65, 80, 85, 90, 95, 99\}$. This ablation confirms that over-activation does compromise the ability to detect OOD data, and ReAct can effectively alleviate this problem. Moreover, when $p$ is sufficiently large, ReAct can improve OOD detection while maintaining a comparable ID classification accuracy. Alternatively, once a sample is detected to be ID, one can always use the original activation $h(\mathbf{x})$, *which is guaranteed to give identical classification accuracy*. When $p$ is too small, OOD performance starts to degrade as expected.

| Rectification percentile | FPR95 ↓ | AUROC ↑ | AUPR ↑ | ID ACC. ↑ | Rectification threshold $c$ |
|---|---|---|---|---|---|
| No ReAct [33] | 58.41 | 86.17 | 96.88 | 75.08 | $\infty$ |
| $p = 99$ | 44.57 | 90.45 | 97.96 | 75.12 | 2.25 |
| $p = 95$ | 35.39 | 92.39 | 98.37 | 74.76 | 1.50 |
| $p = 90$ | 31.43 | 92.95 | 98.50 | 73.75 | 1.00 |
| $p = 85$ | 34.08 | 92.05 | 98.35 | 72.91 | 0.84 |
| $p = 80$ | 41.51 | 89.54 | 97.91 | 71.93 | 0.72 |
| $p = 65$ | 74.62 | 74.14 | 94.39 | 67.14 | 0.50 |
| $p = 10$ | 74.70 | 57.55 | 86.06 | 1.22 | 0.06 |

Table 2: Effect of rectification threshold for inference. Model is trained on ImageNet using ResNet-50 [11]. All numbers are percentages and are averaged over 4 OOD test datasets.

**Effect on other network architectures.** We show that ReAct is effective on a different architecture in Table 1. In particular, we consider a lightweight model MobileNet-v2 [44], which can be suitable for OOD detection in on-device mobile applications. Same as before, we apply ReAct on the output of the penultimate layer, with the rectification threshold chosen based on the 90-th percentile. Our method reduces the FPR95 by **9.18**% compared to the best baseline [31].

**What about applying ReAct on other layers?** Our results suggest that applying ReAct on the penultimate layer is the most effective, since the activation patterns are most distinctive. To see this, we provide the activation and performance study for intermediate layers in Appendix E (see Figure 6 and Table 6). Interestingly, early layers display less distinctive signatures between ID and OOD data. This is expected because neural networks generally capture lower-level features in early layers (such as Gabor filters [61] in layer 1), whose activations can be very similar between ID and OOD. The semantic-level features only emerge as with deeper layers, where ReAct is the most effective.

## 4.2 Evaluation on CIFAR Benchmarks

**Datasets.** We evaluate on CIFAR-10 and CIFAR-100 [27] datasets as in-distribution data, using the standard split with 50,000 training images and 10,000 test images. For OOD data, we consider six common benchmark datasets: `Textures` [5], `SVHN` [37], `Places365` [62], `LSUN-Crop` [60], `LSUN-Resize` [60], and `iSUN` [57].

**Experimental details.** We train a standard ResNet-18 [11] model on in-distribution data. The feature dimension of the penultimate layer is 512. For both CIFAR-10 and CIFAR-100, the models are trained for 100 epochs. The start learning rate is 0.1 and decays by a factor of 10 at epochs 50, 75, and 90. For threshold $c$, we use the 90-th percentile of activations estimated on the ID data.

**ReAct is compatible with various OOD scoring functions.** We show in Table 3 that ReAct is a flexible method that is compatible with alternative scoring functions $S(\mathbf{x}; f^{\text{ReAct}})$. To see this, we consider commonly used scoring functions, and compare the performance both with and without using ReAct respectively. In particular, we consider softmax confidence [15], ODIN score [31] as well as energy score [33]—all of which derive OOD scores directly from the output $f(\mathbf{x})$. In particular, using ReAct on energy score yields the best performance, which is desirable as energy is a hyperparameter-free OOD score and is easy to compute in practice. Note that Mahalanobis [29] estimates OOD score using feature representations instead of the model output $f(\mathbf{x})$, hence is less compatible with ReAct. On all three in-distribution datasets, using ReAct consistently outperforms the counterpart without rectification. Results in Table 3 are based on the average across multiple OOD test datasets. *Detailed performance for each OOD test dataset is provided in Appendix, Table 7.*

| Method | CIFAR-10 | | | CIFAR-100 | | | ImageNet | | |
|---|---|---|---|---|---|---|---|---|---|
| | FPR95 ↓ | AUROC ↑ | AUPR ↑ | FPR95 ↓ | AUROC ↑ | AUPR ↑ | FPR95 ↓ | AUROC ↑ | AUPR ↑ |
| Softmax score [15] | 56.71 | 91.17 | 79.11 | 80.72 | 76.83 | 78.41 | 66.95 | 81.99 | 95.76 |
| Softmax score + ReAct | 53.81 | 91.70 | 92.11 | 75.45 | 80.40 | 84.28 | 58.28 | 87.06 | 97.22 |
| Energy [33] | 35.60 | 93.57 | 95.01 | 71.93 | 82.82 | 86.28 | 58.41 | 86.17 | 96.88 |
| Energy+ReAct | 32.91 | **94.27** | **95.53** | **59.61** | **87.48** | **89.63** | **31.43** | **92.95** | **98.50** |
| ODIN [31] | 31.10 | 93.79 | 94.95 | 66.21 | 82.88 | 86.25 | 56.48 | 85.41 | 96.61 |
| ODIN+ReAct | **28.81** | 94.04 | 94.82 | 59.91 | 85.23 | 87.53 | 44.10 | 90.70 | 98.04 |

Table 3: **Ablation results.** ReAct is compatible with different OOD scoring functions. For each ID dataset, we use the same model and compare the performance with and without ReAct respectively. ↑ indicates larger values are better and ↓ indicates smaller values are better. All values are percentages and are averaged over multiple OOD test datasets. Detailed performance for each OOD test dataset is available in Table 7 in Appendix.

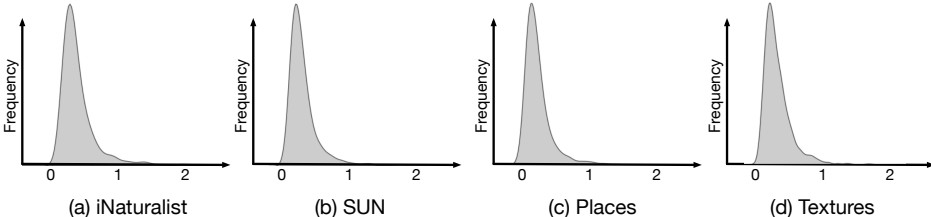

Figure 2: Positively skewed distribution of $\mu_i$ (mean of each unit) in the penultimate layer for four OOD datasets (iNaturalist [16], SUN [55], Places [62], Textures [5]). Model is trained on ImageNet. Note that the right tail has a higher density than the left tail. By left and right tail we refer to the samples that are to the left and right of the *median*, which is close to the mode in the case of skewed distribution.

## 5 Theoretical Analysis

To better understand the effect of ReAct, we mathematically model the ID and OOD activations as rectified Gaussian distributions and derive their respective distributions after applying ReAct. These modeling assumptions are based on the activation statistics observed on ImageNet in Figure 1. In the following analysis, we show that ReAct reduces mean OOD activations more than ID activations since OOD activations are more positively skewed (see Appendix A for derivations).

**ID activations.** Let $h(\mathbf{x}) = (z_1, \ldots, z_m) =: \mathbf{z}$ be the activations for the penultimate layer. We assume each $z_i \sim \mathcal{N}^R(\mu, \sigma_{\text{in}}^2)$ for some $\sigma_{\text{in}} > 0$. Here $\mathcal{N}^R(\mu, \sigma_{\text{in}}^2) = \max(0, \mathcal{N}(\mu, \sigma_{\text{in}}^2))$ denotes the rectified Gaussian distribution, which reflects the fact that activations after ReLU have no negative components. Before truncation with ReAct, the expectation of $z_i$ is given by:

$$\mathbb{E}_{\text{in}}[z_i] = \left[1 - \Phi\left(\frac{-\mu}{\sigma_{\text{in}}}\right)\right] \cdot \mu + \phi\left(\frac{-\mu}{\sigma_{\text{in}}}\right) \cdot \sigma_{\text{in}},$$

where $\Phi$ and $\phi$ denote the cdf and pdf of the standard normal distribution, respectively. After rectification with ReAct, the expectation of $\bar{z}_i = \min(z_i, c)$ is:

$$\mathbb{E}_{\text{in}}[\bar{z}_i] = \left[\Phi\left(\frac{c-\mu}{\sigma_{\text{in}}}\right) - \Phi\left(\frac{-\mu}{\sigma_{\text{in}}}\right)\right] \cdot \mu + \left[1 - \Phi\left(\frac{c-\mu}{\sigma_{\text{in}}}\right)\right] \cdot c + \left[\phi\left(\frac{-\mu}{\sigma_{\text{in}}}\right) - \phi\left(\frac{c-\mu}{\sigma_{\text{in}}}\right)\right] \cdot \sigma_{\text{in}},$$

The reduction in activation after ReAct is:

$$\mathbb{E}_{\text{in}}[z_i - \bar{z}_i] = \phi\left(\frac{c-\mu}{\sigma_{\text{in}}}\right) \cdot \sigma_{\text{in}} - \left[1 - \Phi\left(\frac{c-\mu}{\sigma_{\text{in}}}\right)\right] \cdot (c - \mu) \tag{4}$$

**OOD activations.** We model OOD activations as being generated by a two-stage process: Each OOD distribution defines a set of $\mu_i$'s that represent the mode of the activation distribution for unit $i$, and the activations $z_i$ given $\mu_i$ is represented by $z_i | \mu_i \sim \mathcal{N}^R(\mu_i, \tau^2)$ with $\tau > 0$. For instance, the dark gray line in Figure 1 shows the set $\mu_i$'s on the iNaturalist dataset, and the light gray area depicts the distribution of $z_i | \mu_i$. One commonality across different OOD datasets is that the distribution of $\mu_i$ is *positively skewed*. The assumption of positive skewness is motivated by our observation on real OOD data. Indeed, Figure 2 shows the empirical distribution of $\mu_i$ on an ImageNet pre-trained model for four OOD datasets, all of which display strong positive-skewness, *i.e.*, the right tail has a much higher density than the left tail. This observation is surprisingly consistent across datasets and model architectures. Although a more in-depth understanding of the fundamental cause of positive skewness is important, for this work, we chose to rely on this empirically verifiable assumption and instead focus on analyzing our method ReAct.

Utilizing the positive-skewness property of $\mu_i$, we analyze the distribution of $z_i$ after marginalizing out $\mu_i$, which corresponds to averaging across different $\mu_i$'s induced by various OOD distributions. Let $x_i | \mu_i \sim \mathcal{N}(\mu_i, \tau^2)$ so that $z_i | \mu_i = \max(x_i | \mu_i, 0)$. Since $x_i | \mu_i$ is symmetric and $\mu_i$ is positively-skewed, the marginal distribution of $x_i$ is also positively-skewed[2], which we model with the epsilon-skew-normal (ESN) distribution [35]. Specifically, we assume that $x_i \sim \text{ESN}(\mu, \sigma_{\text{out}}^2, \epsilon)$, which has

---

[2]This can be argued rigorously using Pearson's mode skewness coefficient if the distribution of $\mu_i$ is unimodal.

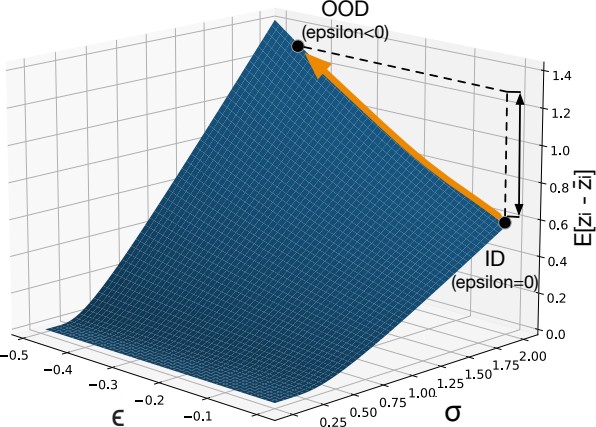

Figure 3: Plot showing the relationship between the skewness parameter $\epsilon$ and the chaotic-ness parameter $\sigma$ on activation reduction after applying ReAct. The function $\mathbb{E}[z_i - \bar{z}_i]$ is increasing in both $-\epsilon$ and $\sigma$, which suggests that ReAct has a greater reduction effect for activation distributions with positive skewness ($\epsilon < 0$) and chaotic-ness—two signature characteristics of OOD activation.

the following density function:

$$q(x) = \begin{cases} \phi((x-\mu)/\sigma_{\text{out}}(1+\epsilon))/\sigma_{\text{out}} & \text{if } x < \mu, \\ \phi((x-\mu)/\sigma_{\text{out}}(1-\epsilon))/\sigma_{\text{out}} & \text{if } x \geq \mu. \end{cases} \tag{5}$$

with $\epsilon \in [-1, 1]$ controlling the skewness. In particular, the ESN distribution is positively-skewed when $\epsilon < 0$. It follows that $z_i = \max(x_i, 0)$, with expectation:

$$\mathbb{E}_{\text{out}}[z_i] = \mu - (1+\epsilon)\Phi\left(\frac{-\mu}{(1+\epsilon)\sigma_{\text{out}}}\right) \cdot \mu + (1+\epsilon)^2 \phi\left(\frac{-\mu}{(1+\epsilon)\sigma_{\text{out}}}\right) \cdot \sigma_{\text{out}} - \frac{4\epsilon}{\sqrt{2\pi}} \cdot \sigma_{\text{out}}. \tag{6}$$

Expectation after applying ReAct becomes:

$$\mathbb{E}_{\text{out}}[\bar{z}_i] = \mu - (1+\epsilon)\Phi\left(\frac{-\mu}{(1+\epsilon)\sigma_{\text{out}}}\right) \cdot \mu + (1-\epsilon)\left[1 - \Phi\left(\frac{c-\mu}{(1-\epsilon)\sigma_{\text{out}}}\right)\right] \cdot (c-\mu)$$

$$+ \left[(1+\epsilon)^2 \phi\left(\frac{-\mu}{(1+\epsilon)\sigma_{\text{out}}}\right) - (1-\epsilon)^2 \phi\left(\frac{c-\mu}{(1-\epsilon)\sigma_{\text{out}}}\right) - \frac{4\epsilon}{\sqrt{2\pi}}\right] \cdot \sigma_{\text{out}}, \tag{7}$$

Hence:

$$\mathbb{E}_{\text{out}}[z_i - \bar{z}_i] = (1-\epsilon)^2 \phi\left(\frac{c-\mu}{(1-\epsilon)\sigma_{\text{out}}}\right) \cdot \sigma_{\text{out}} - (1-\epsilon)\left[1 - \Phi\left(\frac{c-\mu}{(1-\epsilon)\sigma_{\text{out}}}\right)\right] \cdot (c-\mu), \tag{8}$$

which recovers Equation 4 when $\epsilon = 0$ and $\sigma_{\text{out}} = \sigma_{\text{in}}$.

**Remark 1: Activation reduction on OOD is more than ID.** Figure 3 shows a plot of $\mathbb{E}[z_i - \bar{z}_i]$ for $\mu = 0.5$ and $c = 1$. Observe that decreasing $\epsilon$ (more positive-skewness) or increasing $\sigma$ (more chaotic-ness) leads to a larger reduction in the mean activation after applying ReAct. For example, under the same $\sigma$, a larger $\mathbb{E}_{\text{out}}[z_i - \bar{z}_i] - \mathbb{E}_{\text{in}}[z_i - \bar{z}_i]$ can be observed by the gap of $z$-axis value between $\epsilon = 0$ and $\epsilon < 0$ (*e.g.*, $\epsilon = -0.4$). This suggests that rectification on average affects OOD activations more severely compared to ID activations.

**Remark 2: Output reduction on OOD is more than ID.** To derive the effect on the distribution of model output, consider output logits $f(\mathbf{z}) = W\mathbf{z} + \mathbf{b}$ and assume without loss of generality that $W\mathbf{1} > 0$ element-wise. This can be achieved by adding a positive constant to $W$ without changing the output probabilities or classification decision. Let $\delta = \mathbb{E}_{\text{out}}[\mathbf{z} - \bar{\mathbf{z}}] - \mathbb{E}_{\text{in}}[\mathbf{z} - \bar{\mathbf{z}}] > 0$. Then:

$$\mathbb{E}_{\text{out}}[f(\mathbf{z}) - f(\bar{\mathbf{z}})] = \mathbb{E}_{\text{out}}[W(\mathbf{z} - \bar{\mathbf{z}})] = W\mathbb{E}_{\text{out}}[\mathbf{z} - \bar{\mathbf{z}}] = W\left(\mathbb{E}_{\text{in}}[\mathbf{z} - \bar{\mathbf{z}}] + \delta\mathbf{1}\right)$$

$$= \mathbb{E}_{\text{in}}[W(\mathbf{z} - \bar{\mathbf{z}})] + \delta W\mathbf{1}$$

$$> \mathbb{E}_{\text{in}}[f(\mathbf{z}) - f(\bar{\mathbf{z}})].$$

Hence the increased separation between OOD and ID activations transfers to the output space as well. Note that the condition of $W\mathbf{1} > 0$ is sufficient but not necessary for this result to hold. In fact, our experiments in Section 4 do not require this condition. However, we verified empirically that ensuring $W\mathbf{1} > 0$ by adding a positive constant to $W$ and applying ReAct does confer benefits to OOD detection, which validates our theoretical analysis.

**Why ReAct improves the OOD scoring functions?**    Our theoretical analysis above shows that ReAct suppresses logit output for OOD data more so than for ID data. This means that for detection scores depending on the logit output (*e.g.,* energy score [33]), the gap between OOD and ID score will be enlarged after applying ReAct, which makes thresholding more capable of separating OOD and ID inputs; see Figure 1(a) and (c) for a concrete example showing this effect.

## 6   Discussion and Further Analysis

**Why do OOD samples trigger abnormal unit activation patterns?** So far we have shown that OOD data can trigger unit activation patterns that are significantly different from ID data, and that ReAct can effectively alleviate this issue (empirically in Section 4 and theoretically in Section 5). Yet a question left in mystery is *why* such a pattern occurs in modern neural networks? Answering this question requires carefully examining the internal mechanism by which the network is trained and evaluated. Here we provide one plausible explanation for the activation patterns observed in Figure 1, with the hope of shedding light for future research.

Intriguingly, our analysis reveals an important insight that batch normalization (BatchNorm) [22]—a common technique employed during model training—is in fact both a blessing (for ID classification) and a curse (for OOD detection). Specifically, for a unit activation denoted by $z$, the network estimates the running mean $\mathbb{E}_{\text{in}}(z)$ and variance $\text{Var}_{\text{in}}(z)$, over the entire ID training set during training. During inference time, the network applies BatchNorm statistics $\mathbb{E}_{\text{in}}(z)$ and $\text{Var}_{\text{in}}(z)$, which helps normalize the activations for the test data with the same distribution $\mathcal{D}_{\text{in}}$:

$$\text{BatchNorm}(z; \gamma, \beta, \epsilon) = \frac{z - \mathbb{E}_{\text{in}}[z]}{\sqrt{\text{Var}_{\text{in}}[z] + \epsilon}} \cdot \gamma + \beta \tag{9}$$

However, our key observation is that using *mismatched* BatchNorm statistics—that are estimated on $\mathcal{D}_{\text{in}}$ yet blindly applied to the OOD $\mathcal{D}_{\text{out}}$—can trigger abnormally high unit activations. As a thought experiment, we instead apply the *true* BatchNorm statistics estimated on a batch of OOD images and we observe well-behaved activation patterns with near-constant mean and standard deviations—just like the ones observed on the ID data (see Figure 4, top). Our study therefore reveals one of the fundamental causes for neural networks to produce overconfident predictions for OOD data. After applying the true statistics (estimated on OOD), the output distributions between ID and OOD data become much more separable. While this thought experiment has shed some guiding light, the solution of estimating BatchNorm statistics on a batch of OOD data is not at all satisfactory and realistic. Arguably, it poses a strong and impractical assumption of having access to a batch of OOD data during test time. Despite its limitation, we view it as an oracle, which serves as an *upper bound on performance* for ReAct.

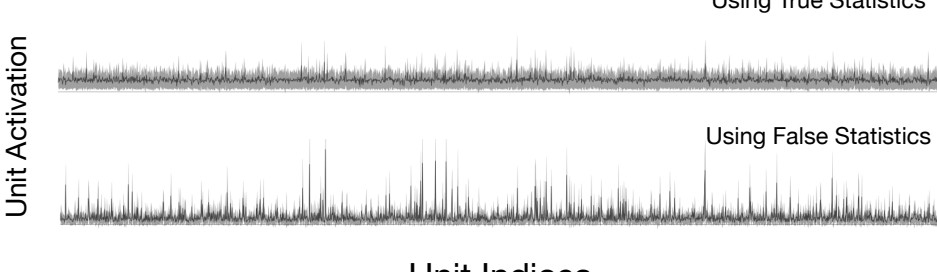

Figure 4: The distribution of per-unit activations in the penultimate layer for OOD data (iNaturalist) by using *true* (top) vs. *mismatched* (bottom) BatchNorm statistics for OOD data.

| Method | iNaturalist | Places | SUN | Textures |
|---|---|---|---|---|
| Oracle (batch OOD for estimating BN statistics) | 99.59 | 99.09 | 98.32 | 91.43 |
| ReAct (single OOD) | 96.22 | 94.20 | 91.58 | 89.80 |
| No ReAct [33] | 89.95 | 85.89 | 82.86 | 85.99 |

Table 4: Comparison with oracle using OOD BN statistics. Model is trained on ImageNet (see Section 4.1). Values are AUROC.

In particular, results in Table 4 suggest that our method favorably matches the oracle performance using the ground truth BN statistics. This is encouraging as our method does not impose any batch assumption and can be feasible for *single-input* testing scenarios.

**What about networks trained with different normalization mechanisms?** Going beyond batch normalization [22], we further investigate (1) whether networks trained with alternative normalization approaches exhibit similar activation patterns, and (2) whether ReAct is helpful there. To answer this question, we additionally evaluate networks trained with WeightNorm [43] and GroupNorm [54]— two other well-known normalization methods. As shown in Figure 5, the unit activations also display highly distinctive signature patterns between ID and OOD data, with more chaos on OOD data. In all cases, the networks are trained to adapt to the ID data, resulting in abnormal activation signatures on OOD data in testing. Unlike BatchNorm, there is no easy oracle (*e.g.*, re-estimating the statistics on OOD data) to counteract the ill-fated normalizations.

We apply ReAct on models trained with WeightNorm and GroupNorm, and report results in Table 5. Our results suggest that ReAct is consistently effective under various normalization schemes. For example, ReAct reduces the average FPR95 by **23.54**% and **14.7**% respectively. Overall, ReAct has shown broad efficacy and compatibility with different OOD scoring functions (Section 4.2).

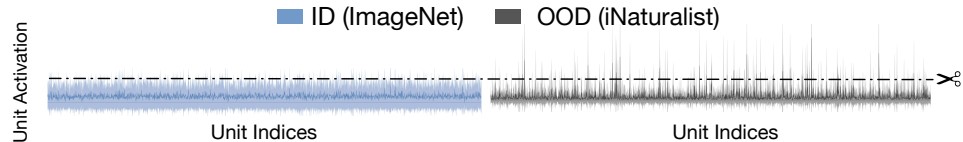

Figure 5: The distribution of per-unit activations in the penultimate layer for ID (ImageNet [7]) and OOD (iNaturalist [16]) on model trained with weight normalization [43].

| | | OOD Datasets | | | | | | | | Average | |
| | Methods | iNaturalist | | SUN | | Places | | Textures | | | |
| | | FPR95 ↓ | AUROC ↑ | FPR95 ↓ | AUROC ↑ | FPR95 ↓ | AUROC ↑ | FPR95 ↓ | AUROC ↑ | FPR95 ↓ | AUROC ↑ |
|---|---|---|---|---|---|---|---|---|---|---|---|
| GroupNorm | w.o. ReAct | 65.38 | 88.45 | 65.11 | 85.52 | 65.46 | 84.34 | 69.17 | 83.22 | 66.28 | 85.38 |
| | w/ ReAct | **39.45** | **92.95** | **51.57** | **87.90** | **52.78** | **87.32** | **62.50** | 81.76 | **51.58** | **87.48** |
| WeightNorm | w.o. ReAct | 40.71 | 92.52 | 48.07 | 89.39 | 50.92 | 87.87 | 61.65 | 80.71 | 50.34 | 87.62 |
| | w/ ReAct | **19.73** | **95.91** | **31.39** | **93.21** | **42.34** | **88.94** | **13.74** | **96.98** | **26.80** | **93.76** |

Table 5: **Effectiveness of ReAct for different normalization methods.** ReAct consistently improves OOD detection performance for the model trained with GroupNorm and WeightNorm. In-distribution is ImageNet-1k dataset. **Bold** numbers are superior results.

## 7  Related Work

**OOD uncertainty estimation with discriminative models.** The problem of classification with rejection can date back to early works on abstention [4, 10], which considered simple model families such as SVMs [6]. A comprehensive survey on OOD detection can be found in [58]. The phenomenon of neural networks' overconfidence in out-of-distribution data is first revealed by Nguyen et al. [38]. Early works attempted to improve the OOD uncertainty estimation by proposing the ODIN score [17, 31], OpenMax score [3], and Mahalanobis distance-based score [29]. Recent work by Liu et al. [33] proposed using an energy score for OOD uncertainty estimation, which can be easily derived from a discriminative classifier and demonstrated advantages over the softmax confidence score both empirically and theoretically. Wang et al. [51] further showed an energy-based approach can improve OOD uncertainty estimation for multi-label classification networks. Huang and Li [18] revealed that approaches developed for common CIFAR benchmarks might not translate effectively into a large-scale ImageNet benchmark, highlighting the need to evaluate OOD uncertainty estimation in a large-scale real-world setting. Previous methods primarily derive OOD scores using original activations. In contrast, our work is motivated by a novel analysis of internal activations, and shows that rectifying activations is a surprisingly effective approach for OOD detection.

**Neural network activation analysis.** Neural networks have been studied at the granularity of the activation of individual layers [21, 34, 59, 63], or individual networks [30]. In particular, Li et al. [30] studied the similarity of activation space between two independently trained neural networks.

Previously, Hein et al. [13] showed that neural networks with ReLU activation can lead to arbitrary high activation for inputs far away from the training data. We show that using ReAct could efficiently alleviate this undesirable phenomenon. ReAct does not rely on auxiliary data and can be conveniently used for pre-trained models. The idea of rectifying unit activation [26], which is known as Relu6, was used to facilitate the learning of sparse features. In this paper, we first show that rectifying activation can drastically alleviate the overconfidence issue for OOD data, and as a result, improve OOD detection.

**OOD uncertainty estimation with generative models.** Alternative approaches for detecting OOD inputs resort to generative models that directly estimate density [8, 20, 23, 41, 49, 50]. An input is deemed as OOD if it lies in the low-likelihood regions. A plethora of literature has emerged to utilize generative models for OOD detection [24, 40, 45, 46, 52, 56]. Interestingly, Nalisnick et al. [36] showed that deep generative models can assign a high likelihood to OOD data. Moreover, generative models can be prohibitively challenging to train and optimize, and the performance can often lag behind the discriminative counterpart. In contrast, our method relies on a discriminative classifier, which is easier to optimize and achieves stronger performance.

**Distributional shifts.** Distributional shifts have attracted increasing research interests [25]. It is important to recognize and differentiate various types of distributional shift problems. Literature in OOD detection is commonly concerned about model reliability and detection of label-space shifts, where the OOD inputs have disjoint labels *w.r.t.* ID data and therefore *should not be predicted by the model*. Meanwhile, some works considered label distribution shift [1, 2, 32, 42, 47, 53], where the label space is common between ID and OOD but the marginal label distribution changes, as well as covariate shift in the input space [14, 39], where inputs can be corruption-shifted or domain-shifted [17, 48]. It is important to note that our work focuses on the detection of shifts where the label space $\mathcal{Y}$ is different between ID and OOD data and hence the model should not make any prediction, instead of label distribution shift and covariate shift where the model is expected to *generalize*.

# 8   Conclusion

This paper provides a simple activation rectification strategy termed ReAct, which truncates the high activations during test time for OOD detection. We provide both empirical and theoretical insights characterizing and explaining the mechanism by which ReAct improves OOD uncertainty estimation. By rectifying the activations, the outsized contribution of hidden units on OOD output can be attenuated, resulting in a stronger separability from ID data. Extensive experiments show ReAct can significantly improve the performance of OOD detection on both common benchmarks and large-scale image classification models. Applications of multi-class classification can benefit from our method, and we anticipate further research in OOD detection to extend this work to tasks beyond image classification. We hope that our insights inspire future research to further examine the internal mechanisms of neural networks for OOD detection.

# 9   Societal Impact

Our project aims to improve the dependability and trustworthiness of modern machine learning models. This stands to benefit a wide range of fields and societal activities. We believe out-of-distribution uncertainty estimation is an increasingly critical component of systems that range from consumer and business applications (*e.g.*, digital content understanding) to transportation (*e.g.*, driver assistance systems and autonomous vehicles), and to health care (*e.g.*, unseen disease identification). Many of these applications require classification models in operation. Through this work and by releasing our code, we hope to provide machine learning researchers with a new methodological perspective and offer machine learning practitioners a plug-and-play tool that renders safety against OOD data in the real world. While we do not anticipate any negative consequences to our work, we hope to continue to build on our method in future work.

# Acknowledgement

YS and YL are supported by the Office of the Vice Chancellor for Research and Graduate Education (OVCRGE) with funding from the Wisconsin Alumni Research Foundation (WARF).

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
