# A   Theoretical Details

Here we derive Equation 8 for $\epsilon \geq 0$ and $\sigma_{\text{out}} = \sigma > 0$. Since $\text{ESN}(\mu, \sigma^2, 0) = \mathcal{N}^R(\mu, \sigma)$, we can obtain Equation 4 for ID activation by specializing the result to $\epsilon = 0$. We begin with a useful lemma.

**Lemma 1.** *Let $X \sim ESN(0, \sigma^2, \epsilon)$ and let $a \leq b \leq 0$, $0 \leq c \leq d$. Then $\mathbb{P}(a \leq X \leq b) = (1 + \epsilon)\left[\Phi\left(\frac{b}{(1+\epsilon)\sigma}\right) - \Phi\left(\frac{a}{(1+\epsilon)\sigma}\right)\right]$ and $\mathbb{P}(c \leq X \leq d) = (1 - \epsilon)\left[\Phi\left(\frac{d}{(1-\epsilon)\sigma}\right) - \Phi\left(\frac{c}{(1-\epsilon)\sigma}\right)\right]$.*

*Proof.*

$$\mathbb{P}(a \leq X \leq b) = (1 + \epsilon)\int_{x=a}^{b} \frac{1}{(1+\epsilon)\sigma} \phi\left(\frac{x}{(1+\epsilon)\sigma}\right) dx$$

$$= (1 + \epsilon)\left[\Phi\left(\frac{b}{(1+\epsilon)\sigma}\right) - \Phi\left(\frac{a}{(1+\epsilon)\sigma}\right)\right]$$

since the integral that of a $\mathcal{N}(0, (1+\epsilon)^2\sigma^2)$ distribution between $a$ and $b$. The result for $\mathbb{P}(c \leq X \leq d)$ follows analogously. $\square$

Suppose that $X_\mu \sim \text{ESN}(\mu, \sigma^2, \epsilon)$ with $\mu > 0$ and let $Z_\mu = \max(X_\mu, 0)$. Define $X = X_\mu - \mu$ so that $Z_\mu = \max(X + \mu, 0) = \max(X, -\mu) + \mu$. We can derive the expectation of $Z := \max(X, -\mu)$:

$$\mathbb{E}[Z] = \underbrace{-\mu \cdot \mathbb{P}(X < -\mu)}_{(I)} + \underbrace{\int_{x=-\mu}^{0} \frac{x}{\sigma}\phi\left(\frac{x}{(1+\epsilon)\sigma}\right) dx}_{(II)} + \underbrace{\int_{x=0}^{\infty} \frac{x}{\sigma}\phi\left(\frac{x}{(1-\epsilon)\sigma}\right) dx}_{(III)}.$$

$$(I) = -\mu(1 + \epsilon)\Phi\left(\frac{-\mu}{(1+\epsilon)\sigma}\right) \quad \text{by Lemma 1;}$$

$$(II) = (1 + \epsilon)\int_{x=-\mu}^{0} \frac{x}{(1+\epsilon)\sigma}\phi\left(\frac{x}{(1+\epsilon)\sigma}\right) dx = (1 + \epsilon)^2\left[\phi\left(\frac{-\mu}{(1+\epsilon)\sigma}\right) - \phi(0)\right]\sigma$$

since the integral is the expectation of an un-normalized truncated Gaussian between $-\mu$ and $0$. Similarly, (III) is $(1 - \epsilon)$ times the expectation of an un-normalized Gaussian between $0$ and $\infty$, thus (III) $= (1 - \epsilon)^2\phi(0)\sigma$. Combining (I)-(III) gives:

$$\mathbb{E}[Z] = -\mu(1 + \epsilon)\Phi\left(\frac{-\mu}{(1+\epsilon)\sigma}\right) + (1 + \epsilon)^2\left[\phi\left(\frac{-\mu}{(1+\epsilon)\sigma}\right) - \phi(0)\right]\sigma + (1 - \epsilon)^2\phi(0)\sigma$$

$$= -\mu(1 + \epsilon)\Phi\left(\frac{-\mu}{(1+\epsilon)\sigma}\right) + (1 + \epsilon)^2\phi\left(\frac{-\mu}{(1+\epsilon)\sigma}\right)\sigma + \phi(0)\sigma[(1 - \epsilon)^2 - (1 + \epsilon)^2]$$

$$= -\mu(1 + \epsilon)\Phi\left(\frac{-\mu}{(1+\epsilon)\sigma}\right) + (1 + \epsilon)^2\phi\left(\frac{-\mu}{(1+\epsilon)\sigma}\right)\sigma - 4\epsilon\phi(0)\sigma$$

$$= -\mu(1 + \epsilon)\Phi\left(\frac{-\mu}{(1+\epsilon)\sigma}\right) + (1 + \epsilon)^2\phi\left(\frac{-\mu}{(1+\epsilon)\sigma}\right)\sigma - \frac{4\epsilon}{\sqrt{2\pi}}\sigma.$$

Equation 6 follows since $\mathbb{E}[Z_\mu] = \mathbb{E}[Z] + \mu$. To derive Equation 7, note that the expectation of $\bar{Z} := \min(Z, c - \mu)$ is given by:

$$\mathbb{E}[\bar{Z}] = (I) + (II) + \underbrace{\int_{x=0}^{c-\mu} \frac{x}{\sigma}\phi\left(\frac{x}{(1-\epsilon)\sigma}\right) dx}_{(IV)} + \underbrace{(c - \mu) \cdot \mathbb{P}(X > c - \mu)}_{(V)}.$$

(IV) $= (1 - \epsilon)^2\left[\phi(0) - \phi\left(\frac{c-\mu}{(1-\epsilon)\sigma}\right)\right]\sigma$ follows from a similar argument as above, and

$$(V) = (c - \mu)(1 - \epsilon)\left[1 - \Phi\left(\frac{c - \mu}{(1+\epsilon)\sigma}\right)\right]$$

can be derived using Lemma 1. Combining (I),(II),(IV),(V) and observing that $\mathbb{E}[\min(Z_\mu, c)] = \mathbb{E}[\bar{Z}] + \mu$ gives Equation 7.

# B  Descriptions of Baseline Methods

For the reader's convenience, we summarize in detail a few common techniques for defining OOD scores that measure the degree of ID-ness on the given sample. All the methods derive the score *post hoc* on neural networks trained with in-distribution data only. By convention, a higher score is indicative of being in-distribution, and vice versa.

**Softmax score**   One of the earliest works on OOD detection considered using the maximum softmax probability (MSP) to distinguish between $\mathcal{D}_{\text{in}}$ and $\mathcal{D}_{\text{out}}$ [15]. In detail, suppose the label space is $\mathcal{Y} = \{1, \ldots, K\}$. We assume the classifier $f$ is defined in terms of a feature extractor $f : \mathcal{X} \to \mathbb{R}^m$ and a linear multinomial regressor with weight matrix $W \in \mathbb{R}^{K \times m}$ and bias vector $\mathbf{b} \in \mathbb{R}^K$. The prediction probability for each class is given by:

$$P(y = k \mid \mathbf{x}) = \text{Softmax}(Wh(\mathbf{x}) + \mathbf{b})_k. \tag{10}$$

The softmax score is defined as $S_{\text{MSP}}(\mathbf{x}; f) := \max_k P(y = k \mid \mathbf{x})$.

**ODIN score**  Liang et al. [31] introduced the ODIN score which incorporates temperature scaling and input preprocessing to improve the separation of MSP for ID and OOD data.

$$P(y = k \mid \tilde{\mathbf{x}}) = \text{Softmax}[(Wh(\tilde{\mathbf{x}}) + \mathbf{b})/T]_k, \tag{11}$$

where $\tilde{\mathbf{x}}$ is the perturbed input. The ODIN score is defined as $S_{\text{ODIN}}(\mathbf{x}; f) := \max_k P(y = k \mid \tilde{\mathbf{x}})$.

**Energy score**   The energy function [33] maps the logit outputs to a scalar $S_{\text{Energy}}(\mathbf{x}; f) \in \mathbb{R}$, which is relatively lower for ID data:

$$S_{\text{Energy}}(\mathbf{x}; f) = -\log \sum_{k=1}^{K} \exp(\mathbf{w}_i^\top h(\mathbf{x}) + b_i). \tag{12}$$

Note that Liu et al. [33] used the *negative energy score* for OOD detection, in order to align with the convention that $S(\mathbf{x}; f)$ is higher for ID data and vice versa.

**Mahalanobis distance**   All the scores above operate on $K$-dimensional outputs of the multinomial regressor. In [29], the authors instead model the feature-level distribution as a mixture of class conditional Gaussians, and propose the *Mahalanobis distance-based score*:

$$S_{\text{Mahalanobis}}(\mathbf{x}; h) := \max_k -(h(\mathbf{x}) - \hat{\mu}_k)^\top \hat{\Sigma}(h(\mathbf{x}) - \hat{\mu}_k), \tag{13}$$

where $\hat{\mu}_k$ is the estimated feature vector mean for class $k$ and $\hat{\Sigma}$ is the estimated covariance matrix (tied across classes).

# C  Selected Categories in OOD Datasets

We use the following list of concepts as OOD test data, selected from the iNaturalist [16], SUN [55], and Places365 [61] datasets. The concepts are curated to be disjoint from the ImageNet-1k labels [18]. For Textures [5], we use the entire dataset.

**iNaturalist**   *Coprosma lucida, Cucurbita foetidissima, Mitella diphylla, Selaginella bigelovii, Toxicodendron vernix, Rumex obtusifolius, Ceratophyllum demersum, Streptopus amplexifolius, Portulaca oleracea, Cynodon dactylon, Agave lechuguilla, Pennantia corymbosa, Sapindus saponaria, Prunus serotina, Chondracanthus exasperatus, Sambucus racemosa, Polypodium vulgare, Rhus integrifolia, Woodwardia areolata, Epifagus virginiana, Rubus idaeus, Croton setiger, Mammillaria dioica, Opuntia littoralis, Cercis canadensis, Psidium guajava, Asclepias exaltata, Linaria purpurea, Ferocactus wislizeni, Briza minor, Arbutus menziesii, Corylus americana, Pleopeltis polypodioides, Myoporum laetum, Persea americana, Avena fatua, Blechnum discolor, Physocarpus capitatus, Ungnadia speciosa, Cercocarpus betuloides, Arisaema dracontium, Juniperus californica, Euphorbia prostrata, Leptopteris hymenophylloides, Arum italicum, Raphanus sativus, Myrsine australis, Lupinus stiversii, Pinus echinata, Geum macrophyllum, Ripogonum scandens, Echinocereus triglochidiatus, Cupressus macrocarpa, Ulmus crassifolia, Phormium tenax, Aptenia cordifolia, Osmunda claytoniana, Datura*

*wrightii, Solanum rostratum, Viola adunca, Toxicodendron diversilobum, Viola sororia, Uropappus lindleyi, Veronica chamaedrys, Adenocaulon bicolor, Clintonia uniflora, Cirsium scariosum, Arum maculatum, Taraxacum officinale officinale, Orthilia secunda, Eryngium yuccifolium, Diodia virginiana, Cuscuta gronovii, Sisyrinchium montanum, Lotus corniculatus, Lamium purpureum, Ranunculus repens, Hirschfeldia incana, Phlox divaricata laphamii, Lilium martagon, Clarkia purpurea, Hibiscus moscheutos, Polanisia dodecandra, Fallugia paradoxa, Oenothera rosea, Proboscidea louisianica, Packera glabella, Impatiens parviflora, Glaucium flavum, Cirsium andersonii, Heliopsis helianthoides, Hesperis matronalis, Callirhoe pedata, Crocosmia × crocosmiiflora, Calochortus albus, Nuttallanthus canadensis, Argemone albiflora, Eriogonum fasciculatum, Pyrrhopappus pauciflorus, Zantedeschia aethiopica, Melilotus officinalis, Peritoma arborea, Sisyrinchium bellum, Lobelia siphilitica, Sorghastrum nutans, Typha domingensis, Rubus laciniatus, Dichelostemma congestum, Chimaphila maculata, Echinocactus texensis*

**SUN** *badlands, bamboo forest, bayou, botanical garden, canal (natural), canal (urban), catacomb, cavern (indoor), cornfield, creek, crevasse, desert (sand), desert (vegetation), field (cultivated), field (wild), fishpond, forest (broadleaf), forest (needle leaf), forest path, forest road, hayfield, ice floe, ice shelf, iceberg, islet, marsh, ocean, orchard, pond, rainforest, rice paddy, river, rock arch, sky, snowfield, swamp, tree farm, trench, vineyard, waterfall (block), waterfall (fan), waterfall (plunge), wave, wheat field, herb garden, putting green, ski slope, topiary garden, vegetable garden, formal garden*

**Places** *badlands, bamboo forest, canal (natural), canal (urban), cornfield, creek, crevasse, desert (sand), desert (vegetation), desert road, field (cultivated), field (wild), field road, forest (broadleaf), forest path, forest road, formal garden, glacier, grotto, hayfield, ice floe, ice shelf, iceberg, igloo, islet, japanese garden, lagoon, lawn, marsh, ocean, orchard, pond, rainforest, rice paddy, river, rock arch, ski slope, sky, snowfield, swamp, swimming hole, topiary garden, tree farm, trench, tundra, underwater (ocean deep), vegetable garden, waterfall, wave, wheat field*

# D   Hardware

All experiments are conducted on NVIDIA GeForce RTX 2080Ti GPUs.

# E   Ablation Study on Different Layers

We provide the activation patterns for intermediate layers in Figure 6 and the OOD detection performance of applying ReAct to these layers in Table 6. In particular, there are four residual blocks in the original ResNet-50 network [12]. The four layers (denoted by `layer 1` - `layer 4`) are taken from the output of each residual block. Interestingly, early layers display less distinctive signatures between ID and OOD data and ReAct performs worse than the baseline [33] when it is applied on `layer 1` - `layer 3`. This is expected because neural networks generally capture lower-level features in early layers (such as Gabor filters [60]), whose activations can be very similar between ID and OOD. The semantic-level features only emerge as with deeper layers, where ReAct is the most effective.

| Layers of applying ReAct | ID: CIFAR-100 | | ID: ImageNet | |
|---|---|---|---|---|
| | FPR95 ↓ | AUROC ↑ | FPR95 ↓ | AUROC ↑ |
| Layer1 | 90.86 | 68.17 | 84.83 | 74.88 |
| Layer2 | 84.12 | 75.32 | 76.25 | 79.37 |
| Layer3 | 73.4 | 80.91 | 63.87 | 86.46 |
| Layer4 (ReAct) | **59.61** | **87.48** | **31.43** | **92.95** |
| No ReAct [33] | 71.93 | 82.82 | 58.41 | 86.17 |

Table 6:   Ablation study of applying ReAct on different layers. We used ResNet-18 [11] pre-trained on CIFAR-100 and ResNet-50 pre-trained on ImageNet. ↑ indicates larger values are better and ↓ indicates smaller values are better. All values are percentaged over multiple OOD test datasets described in Section 4.2 and Section 4.1.

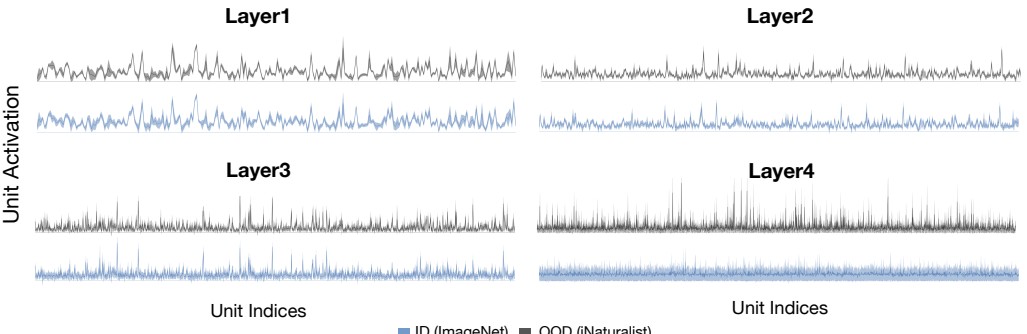

Figure 6: The distribution of per-unit activations in the penultimate layer for ID (ImageNet, blue) and OOD (iNaturalist, gray) on different layers. Each layer (denoted by `layer 1` - `layer 4`) corresponds to the output of each residual block in ResNet-50 [12].

# F  Using True BatchNorm Statistics on OOD Data

Typically, for a unit activation denoted by $z$, the network estimates the running mean $\mathbb{E}_{\text{in}}(z)$ and variance $\text{Var}_{\text{in}}(z)$, over the entire ID training set during training. During inference time, the network applies BatchNorm statistics [22] $\mathbb{E}_{\text{in}}(z)$ and $\text{Var}_{\text{in}}(z)$, which helps normalize the activations for the test data with the same distribution $\mathcal{D}_{\text{in}}$:

$$\text{BatchNorm}(z; \gamma, \beta, \epsilon) = \frac{z - \mathbb{E}_{\text{in}}[z]}{\sqrt{\text{Var}_{\text{in}}[z] + \epsilon}} \cdot \gamma + \beta \tag{14}$$

However, our key observation is that using *mismatched* BatchNorm statistics—that are estimated on $\mathcal{D}_{\text{in}}$ yet blindly applied to the OOD $\mathcal{D}_{\text{out}}$—can trigger abnormally high unit activations (see bottom of Figure 4). As a thought experiment, for OOD data, we instead apply the *true* BatchNorm statistics estimated on a batch of OOD images:

$$\text{BatchNorm}(z; \gamma, \beta, \epsilon) = \frac{z - \mathbb{E}_{\text{out}}[z]}{\sqrt{\text{Var}_{\text{out}}[z] + \epsilon}} \cdot \gamma + \beta. \tag{15}$$

As a result, we observe well-behaved activation patterns with near-constant mean and standard deviations (see the top of Figure 4). Our study therefore reveals one of the fundamental causes for neural networks to produce overconfident predictions for OOD data. Despite the interesting observation, we note that estimating the true BN statistics for OOD poses a strong and impractical assumption of having access to a batch of OOD data during test time. In contrast, using ReAct does not operate under such an assumption and can be applied on any single OOD instance, as well as for neural networks trained with alternative normalization mechanisms (as we show in Section 6).

# G  Unit Activation Patterns for Gaussian Noise

We provide the activation patterns for Gaussian noise input (our validation data) in Figure 7. The experiment is based on ResNet-50 architecture [12]. We show that using Gaussian noise as input can lead to overly high unit activations, which is consistent with the observation in Figure 1 and Figure 5.

| ID Dataset | Methods | SVHN | LSUN-Crop | LSUN-Resize | iSUN | Textures | Places365 | Average |
|---|---|---|---|---|---|---|---|---|
| | | | | FPR95↓/AUROC↑/AUPR↑ | | | | |
| CIFAR-10 | Softmax score [15] | 59.66/91.25/78.84 | 45.21/93.80/80.81 | 51.93/92.73/80.04 | 54.57/92.12/80.01 | 66.45/88.5/79.47 | 62.46/88.64/75.48 | 56.71/91.17/79.11 |
| | Softmax score + ReAct | 57.15/91.69/91.77 | 46.37/93.33/93.00 | 46.32/93.61/93.37 | 50.02/92.96/93.26 | 62.85/89.31/92.59 | 60.15/89.28/88.65 | 53.81/91.70/92.11 |
| | ODIN [31] | 60.37/88.27/89.82 | 7.81/98.58/98.73 | 9.24/98.25/98.51 | 11.62/97.91/98.38 | 52.09/89.17/93.72 | 45.49/90.58/90.55 | 31.10/93.79/94.95 |
| | ODIN+ReAct | 51.77/88.87/89.09 | 14.99/97.29/97.42 | 6.84/98.65/98.81 | 9.55/98.28/98.62 | 43.81/90.41/94.16 | 45.87/90.73/90.82 | 28.81/94.04/94.82 |
| | Energy [33] | 54.41/91.22/93.05 | 10.19/98.05/98.33 | 23.45/96.14/96.92 | 27.52/95.59/96.78 | 55.23/89.37/94.01 | 42.77/91.02/90.98 | 35.60/93.57/95.01 |
| | Energy+ReAct | 49.77/92.18/93.67 | 16.99/97.11/97.48 | 17.94/96.98/97.56 | 20.84/96.46/97.38 | 47.96/91.55/95.40 | 43.97/91.33/91.66 | 32.91/94.27/95.53 |
| CIFAR-100 | Softmax score [15] | 81.32/77.74/78.78 | 70.11/83.51/83.02 | 82.46/75.73/76.32 | 82.26/76.16/78.26 | 85.11/73.36/80.79 | 83.06/74.47/73.27 | 80.72/76.83/78.41 |
| | Softmax score + ReAct | 74.17/82.3/85.58 | 73.1/82.47/85.25 | 74.73/80.81/83.77 | 73.49/81.45/85.60 | 74.82/80.37/89.03 | 82.37/74.99/76.43 | 75.45/80.4/84.28 |
| | ODIN [31] | 40.94/93.29/94.49 | 28.72/94.51/94.93 | 79.61/82.13/85.09 | 76.66/83.51/87.35 | 83.63/72.37/82.80 | 87.71/71.46/72.85 | 66.21/82.88/86.25 |
| | ODIN+ReAct | 22.87/95.63/96.13 | 36.61/91.45/91.20 | 75.02/85.53/88.42 | 70.21/86.51/89.89 | 66.79/82.69/89.52 | 87.94/69.57/70.01 | 59.91/85.23/87.53 |
| | Energy [33] | 81.74/84.56/88.39 | 34.78/93.93/94.77 | 73.57/82.99/85.57 | 73.36/83.80/87.40 | 85.87/74.94/84.12 | 82.23/76.68/77.40 | 71.93/82.82/86.28 |
| | Energy+ReAct | 70.81/88.24/91.07 | 39.99/92.51/93.38 | 54.47/89.56/91.07 | 51.89/90.12/92.29 | 59.15/87.96/93.31 | 81.33/76.49/76.63 | 59.61/87.48/89.63 |

Table 7: Detailed results on six common OOD benchmark datasets: Textures [5], SVHN [37], Places365 [61], LSUN-Crop [59], LSUN-Resize [59], and iSUN [57]. For each ID dataset, we use the same ResNet-18 architecture [11] and compare the performance with and without ReAct respectively. ↑ indicates larger values are better and ↓ indicates smaller values are better.

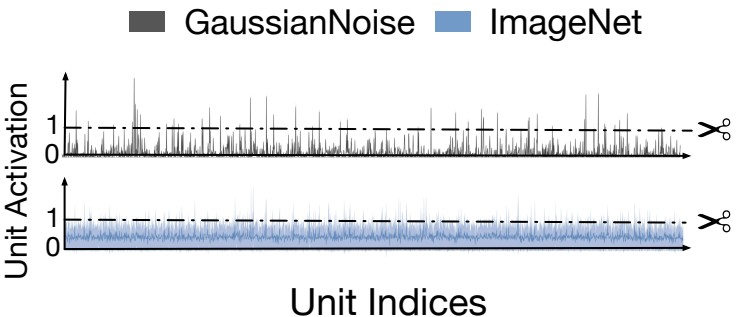

Figure 7: The distribution of per-unit activations in the penultimate layer for OOD data (Gaussian Noise) and ID data (ImageNet) on ResNet50.