# OpenReview forum: "ReAct: Out-of-distribution Detection With Rectified Activations"
_NeurIPS.cc/2021/Conference — NeurIPS 2021 Poster_

### Official Review · Reviewer_QB7D · 2021-06-26

**Rating:** 7
**Confidence:** 4

**Summary:**

This paper identifies that out-of-distribution (OOD) data tend to cause overconfident activations much more often than in-distribution data. Based on this observation, a simple approach called ReAct (Rectified Activations) is proposed, which essentially applies a threshold to the activations of the penultimate layer of a CNN classifier. Extensive experiments demonstrate that ReAct works with different network architectures and OOD detection algorithms, and the improvements over previous SOTA methods are significant.

**Main Review:**

**Strength**

- ReAct is well motivated. Though the design is simple, it works very well with different network architectures and is compatible with several existing OOD detection methods.
- A theoretical analysis is provided to justify ReAct's effect on reducing both the activations and output for OOD data.
- The experiments are extensive and the results are strong.
- The analysis part is insightful, especially the discussions that connect the overconfidence issue to BatchNorm. I think the findings and results could be useful to research beyond OOD detection, such as OOD generalization and domain adaptation.
- The paper is well written and easy to follow.

**Weakness**

I do not find major issues in this paper. The following comments are mainly related to the presentation.

- By just reading the abstract and the introduction, it is unclear how and why ReAct helps OOD detection. It would be clearer to mention that the OOD detection part is based on existing methods such as softmax scores or the energy-based scores. It would also be better to highlight that the idea of ReAct generalizes to different network architectures and works with different OOD detection methods.
- Though the motivation for trimming the activations is intuitive, the reason why ReAct improves the OOD scoring functions is a bit vague. Taking the energy function (Liu et al., 2020) as example, which essentially associates the sum of logits to the marginal distribution (the likelihood). It is less intuitive why lowering the confidence on OOD samples helps make the OOD scores more distinguishable.
- Line199: Fig.2 shows that the left tail has a higher frequency than the right tail. Why the right tail has a higher density? Is this a typo or is there something wrong with my understanding? Please clarify on this.

---
Post-rebuttal update:

I have read the authors' responses and other reviewers' comments.

Overall, I find the proposed approach simple and well-motivated, and the experiments solid and extensive enough for supporting the claim. Hence, I think the paper is worth sharing with the community and keep my rating unchanged.

**Time Spent Reviewing:**

3

---

> ### Author Response · Authors · 2021-08-10
> **Thank you for the positive and constructive feedback**
>
> We thank the reviewer for the positive comments! We are more than _encouraged_ that you found our method simple, well-motivated, extensively studied, and insightful. We certainly share the excitement that our theoretical findings (especially the connection between overconfidence and BatchNorm) can have implications for work beyond OOD detection.
>
> **1. Suggestion on the abstract and introduction**
> Great suggestion! We have revised our abstract and introduction to highlight the basis of the scoring functions, as well as the fact that ReAct generalizes to different network architectures and works with different OOD detection methods.
>
> **2. Why ReAct improves the OOD scoring functions**
> As you pointed out in your review, our theoretical analysis in Section 5 shows that ReAct suppresses logit output for OOD data more so than for ID data. This means that for detection scores such as energy score, the gap between OOD and ID score will be enlarged after applying ReAct, which makes thresholding more capable of separating OOD and ID inputs; see Figure 1(a) and (c) for a concrete example showing this effect. We have clarified this in the updated draft.
>
> **3. Why does the right tail have a higher density?**
> By left and right tail we refer to the samples that are to the left and right of the median, which in the case of skewed distribution is close to the mode; please see https://en.wikipedia.org/wiki/Skewness for an illustration. It is evident that there are fewer points that are far away on the left side of the median than on the right side, hence a higher density for the right tail. Thank you for raising this question---we will add a median line to these plots to help with the visualization.

---

### Official Review · Reviewer_hJUX · 2021-07-13

**Rating:** 4
**Confidence:** 4

**Summary:**

The paper proposes a very simple way of improving out-of-distribution detection performance in neural network classifiers by rectifying the pre-logit layer's activations.


**Limitations And Societal Impact:**

The paper omits the mandatory discussion of their paper's weaknesses and limitations. No reason for this is given in the checklist.

**Main Review:**

Strengths:
The approach is very simple to apply.
It is good that the authors demonstrate results on ImageNet which is not standard in the field of OOD detection.

Weaknesses:
The authors claim to compare to state-of-the-art methods, but the highest performing methods in the literature are variations on outlier exposure [16]. The authors correctly point out that this method uses additional data at train time. I think the authors should include separate evaluations, with their method being applied to outlier-aware approaches (like [16]) in addition to the outlier-unaware approaches that they study.

In the related work section the authors bring up [14] as an example of using an "auxiliary OOD dataset" during training. This seems like a stretch since that paper uses a noise distribution generated from the in-distribution as training out-distribution. By the same logic one could claim that this paper uses an "auxiliary OOD dataset" because they use a noise dataset as validation set for hyperparameter selection.

The technical contribution by the paper is trivial. Also, as the authors mention, the idea of clipping activations has been around for over ten years (clippedReluLayer even having an official implementation in matlab...), so the novelty of the approach is extremely limited.

The paper should definitely include their OOD performance measures for each test out-distribution (as opposed to just the means) in the appendix.

The authors show that their method leads to a degradation in test accuracy (which they correctly point out can be mitigated by an additional forward pass). However, it would be interesting to know if this drop in performance still occurs when training or even just fine-tuning with the ReAct activation. In general, I don't see why the paper doesn't investigate if doing this changes their performance at all.


Clarity:
The paper is mostly simple to follow. However, there are a few issues.
In line 68 f is defined as a function from X to Y. However, later (e.g. in Eq. 2) it is clearly assumed that f represents logits and that f is a function onto R^|Y|.

It is not entirely obvious how the percentile is chosen. Line 120 states "We use a validation set of Gaussian noise" so presumably this is the OOD dataset that is used to choose the percentile. However, first of all no details of this Gaussian noise are given (mean, variance, clip to image space or not?) and the authors do not even state which metric is being optimized for (AUROC, FPR95, AUPR?).


Additional feedback:
In line 243 the authors state that it is surprising that using OOD batch norm statistics normalizes the mean and variance of the activations. I don't see how this is surprising at all. That is the point of batch norm after all.

The writing of the paper needs much improvement. The style is occasionally too informal (e.g. line 22 "[...] modern neural networks can, contrary to popular belief, produce overconfident prediction [...]", who holds this "popular belief"?) and there are many typos (e.g. line 84 an->a, line 115 class->classes, line 140 given->give etc...).

The formatting of paragraphs is somewhat inconsistent. Some end with periods ".", others do not.

**Time Spent Reviewing:**

4

---

> ### Author Response · Authors · 2021-08-10
> **Thank you for the feedback shared; please see clarifications in our response**
>
> We are glad that the reviewer found our method simple to use and broadly evaluated on multiple datasets. Below we address the feedback and comments in detail:
>
> **1. Clarification on the method scope**
> We acknowledge several lines of works in OOD detection literature, including post-hoc and regularization-based methods. Our specific focus is to improve post-hoc OOD detection methods, i.e., based on neural networks trained for classification without outlier regularization.
> We see our contribution to the field complementary (and perhaps orthogonal) to regularization-based methods. We provide two reasons:
>
>
> + First, post-hoc methods have the advantages of easy to use and general applicability without modifying the training procedure and objective. The latter property is especially important for the adoption of OOD detection methods in real-world production environments, where the overhead cost of retraining can be prohibitive.
>
> + Second, the overconfidence issue of OOD data is **technically more challenging** in the post-hoc setting, **particularly due to the lack of outlier data and regularization during training time**. There have been years of effort in the research community to make improvements, including many baselines we compared to. With the regularization and auxiliary outlier data in place, one would do better to reduce the overconfidence issue (as already demonstrated in the related papers). This is because when training with regularization, the auxiliary outlier data’s mean and standard deviations are already considered in the BN and model’s optimization process. The OOD data will display less overly high activations, as opposed to pre-trained models without outlier regularization. For this reason, the motivation and targeting problem of ReACT is no longer relevant here. We have clarified this and added discussion in our updated draft.
>
>
> + For fairness and clarity, we already distinguished our scope of comparison in several places (see **L55, L78, the caption of Table 1, L124, L131**). Importantly, we showed ReAct can work with different post-hoc OOD detection methods and show even stronger performance on already existing competitive baselines (see **Table 3**). We have also revised our SOTA claim being in the post-hoc method family.
>
>
> **2. Clarification on auxiliary OOD**
> + First, the citation of [14] appears to be misplaced - thank you for pointing that out! We meant to cite [14] in the following paragraph _“In particular, [14] showed that neural networks with ReLU activation can lead to arbitrary high activation for inputs far away from the training data.”_ We believe [14] is more advantageous than using OE, without using an auxiliary image dataset. We apologize for the confusion here and have revised the related work section.
>
> + Second, we would also like to mention the key difference between using additional data during training time vs. validation. As discussed above, when training with additional data, its mean and standard deviations are already considered in the BN and optimization process and hence the model’s parameterization. This will change accordingly how the network responds to OOD data. In contrast, in a pre-trained neural network,   the model’s parameters are not affected by the additional data.
>
> + Third, to better understand the role of training-time regularization, we analyzed the model from training-time regularization such as [14]. Given its effective regularization, we observed that it helps produce less overly high activations on OOD data. For this reason, the motivation and targeting problem of ReACT is not relevant anymore. We have clarified this and added discussion in our updated draft. Note that the OE method is no longer reproducible, as the auxiliary dataset TinyImages has been officially taken down by the owners (https://groups.csail.mit.edu/vision/TinyImages/).
>
> **3. Clarification on ID accuracy**
>
> As discussed above, we chose to focus on the post-hoc setting for its ease of use and general applicability. We additionally verified using ReAct during training time, and the resulting accuracy is almost the same as using ReAct post-hoc. Moreover, as the reviewer recognized, one can always guarantee the same accuracy by using the original activation without rectification. For these reasons, we believe our original method is more advantageous, given its overall convenience & performance guarantee.
>
> | Methods       | CIFAR-10 | CIFAR-100 |
> |---------------|----------|-------|
> | ReACT (post-hoc)         | 94.21    | 74.58 |
> | ReACT (train) | 94.03    | 74.6 |
>
>
> **4. Technical contribution**
> We respectfully disagree. We believe every research should be evaluated in the proper context and problem scope. **Our work is the first to investigate the effect of activation rectification for OOD detection, which shows substantial improvement over previous SOTA post-hoc methods. Extensive empirical (Section 4) and theoretical insights (Section 5 & 6) are entirely new contributions to the field.** The novelty and significance of our results are endorsed by all other reviewers. In particular, as reviewer 4 recognizes, _“the analysis part is insightful, especially the discussions that connect the overconfidence issue to BatchNorm. I think the findings and results could be useful to research beyond OOD detection, such as OOD generalization and domain adaptation”_.
>
> The paper by Krizhevsky in 2010 trained a two-layer deep belief network, and the ReLU unit intends to learn sparse features. We do not see any resemblance in terms of both problem formulation or experimental setting. As far as we know, the problem of OOD detection for neural networks was not established back then.
>
>
> **5. OOD performance measures for each test out-distribution**
> We agree on the importance of providing performance figures on the individual OOD test datasets, which is already shown in **Tables 1 and 4**. The performance for CIFAR has been added to the appendix for completeness.
>
>
> **6. Details of Gaussian noise**
> Great suggestion. We sample from $N(0,1)$ for each pixel location. And we used FPR95 as the metric to be optimized for. Details have been added to the main paper for clarity.
>
> **7. Writing glitches**
> All fixed. Thank you for pointing them out!

---

> > ### Comment · Reviewer_hJUX · 2021-08-31
> > **Further clarification**
> >
> > I thank the authors for taking the time to respond to most of my concerns. However, there is one issue that I would still like clarification on.
> > In my review, I wrote: "In line 243 the authors state that it is surprising that using OOD batch norm statistics normalizes the mean and variance of the activations. I don't see how this is surprising at all. That is the point of batch norm after all." I believe that you did not address this concern of mine. Could you please help clear up my confusion?
> >
> > Concretely, you write in the paper: **"As a thought experiment, we instead apply the true BatchNorm statistics estimated on a batch of OOD images, and surprisingly, we observe well-behaved activation patterns with near-constant mean and standard deviations—just like the ones observed on the ID data."** By the definition of batch norm this behavior must hold trivially. The way I see it, this observation is central to your theoretical insight, but also completely obvious. This is why I believed the theoretical contribution to be rather limited. What am I missing here?

---

> > > ### Author Response · Authors · 2021-08-31
> > > **response**
> > >
> > > We thank you for the follow-up. Glad to hear that our response cleared your concerns. We are happy to further clarify misunderstandings regarding the theoretical contributions.
> > >
> > > Our **central theoretical contributions are in Section 5**, which are entirely new and directly support our empirical results in Section 4. To recap our theory contributions:
> > > - Contribution 1: We mathematically model the ID and OOD activations as rectified Gaussian distributions and derive their respective distributions after applying ReAct (**L185-L210**).
> > > - Contribution 2: We show that ReAct reduces mean OOD activations more than ID activations, and benefits OOD detection performance (**L211-L225**).
> > > - Rigorous proofs are given in the Appendix to support theoretical claims.
> > >
> > > > _The way I see it, this observation is central to your theoretical insight_
> > >
> > > In lines L241-243, we did not claim this to be central to our theoretical contribution. **Our theory contribution in Section 5 does not depend on observation in Section 6, and holds regardless of whether BN is the culprit of the OOD activation distribution.** We see Section 6 complementary to our theoretical contributions in Section 5. As we specified as the motivation of Section 6: “Here we provide one plausible explanation for the activation patterns observed in Figure 1, with the hope of shedding light for future research”.
> > >
> > > To recap our new contributions in Section 6:
> > > - We provide one explanation for the cause of the activation patterns observed for OOD data.
> > > - We verified our hypothesis experimentally by applying BN statistics of OOD data. We view it as an upper bound performance of ReACT. The purpose of our analysis is clearly stated in **L246-L250**.
> > > - We showed the contribution of ReACT goes beyond batch normalization. We show that ReAct is effective under various normalization schemes, including GroupNorm and WeightNorm as well.
> > >
> > > What we intended to highlight is the subsequent clause after L243---that in a hypothetical world where OOD BN statistics are available, applying the BN statistics of OOD data makes the ID/OOD well separated. While the behavior may be expected in retrospect, establishing the link of the OOD activation pattern from a BN perspective is not previously obvious to the field of OOD detection. Nevertheless, we do recognize that “surprising” or “not surprising at all” are subjective terms, and have removed the wording to keep our claims and descriptions of findings factual. Thank you for pointing that out.

---

> > > > ### Comment · Reviewer_hJUX · 2021-09-10
> > > > **response**
> > > >
> > > > I thank the authors for their response.
> > > > I believe my initial score was too low and I have raised it.

---

### Official Review · Reviewer_68cE · 2021-07-16

**Rating:** 6
**Confidence:** 3

**Summary:**

The authors propose a simple method for OOD detection, called ReAct, which truncates high activations at test time. Experiments show that ReAct performs better than state-of-the-art OOD detection methods. The authors also theoretically verify the effectiveness of ReAct.

**Limitations And Societal Impact:**

The authors claim that the proposed method shows good performance on a variety of data and settings and can be combined with various existing OOD methods, but they are not very explicit about the limitations of the proposed method. For example, this study proves that the proposed method is effective under the hypothesis that the distribution of OOD is positively skewed, but if this does not hold, there should be no guarantee that the proposed method is effective. The authors should include such discussions.

**Main Review:**

The proposed ReAct is simple, novel, and shows high performance compared to conventional OOD detection methods. In addition, the entire paper is very readable with clear arguments. However, there are some inadequacies in the verification and other aspects as follows.

- I think that the performance of ReAct is greatly affected by the layer at which it is applied, but this is not fully verified in this paper. The Appendix shows the differences in activation for each layer, suggesting that the deeper the layer, the easier it is to distinguish between ID and OOD. However, the authors do not indicate the extent to which these differences actually affect performance. In addition, the Appendix examines the activity values of ResNet-50, but they should verify and show whether the trend is similar for other architectures (such as ResNet-18 used in Sec. 4.2).
- In the proof, the authors model the distribution of OOD as the epsilon skew-normal distribution, since it is positively skewed. In the proof, the distribution of OOD is positively skewed, so it is modeled by the epsilon skew-normal distribution. Under this assumption, they prove that ReAct is effective. However, it does not explain why the distribution of OOD becomes positively skewed. In addition, it is not clear whether there is a case where the distribution of OOD is not positively skewed (which is directly related to a limitation of the proposed method). The authors should clarify them.
- According to the setup, the authors use a validation set of Gaussian noise to determine the value of p. However, in order for this Gaussian noise tuned p to be valid for testing, the validation set of Gaussian noise and the test set, i.e., the OOD, should have the same distributional trend. The authors should verify this.
- Other minor comments
  - This method is simple and should be easy to implement, but I think it would help the reader to understand it better if the code is included.
  - There should be a clear formulation of what S is around Equation 3.  I found out later that this is a score function, meaning any existing OOD scoring function, but it should not be introduced into Equation 3 without explaining.

**Time Spent Reviewing:**

3

---

> ### Author Response · Authors · 2021-08-10
> **Thank you for the encouraging and constructive feedback**
>
> We are encouraged that the reviewer found our method simple, novel, effective, and clearly presented. We particularly appreciate the reviewer for the constructive feedback. We address the reviewer’s comments below, which helped us greatly strengthen the manuscript.
>
> **1. Effect of different layers**
> Great point! We agree it’s better to provide numerical results on the performance of varying layers; see attached below. We have also updated the appendix with the new Table. We have also verified the trend of activation patterns on ResNet-18, which is indeed similar. Additional plots have been added to the appendix.
>
> CIFAR-100
>
> | Layers of applying ReAct | FPR95 | AUROC |
> | ------ | ------ | ------ |
> | Layer1 | 90.86 |  68.17 |
> | Layer2 | 84.12 | 75.32 |
> | Layer3 | 73.40 |   80.91 |
> | Layer4 (ReAct) | **59.61** |   **87.48** |
> | No ReAct | 71.93 |   82.82   |
>
> ImageNet
>
> | Layers of applying ReAct | FPR95 | AUROC |
> | ------ | ------ | ------ |
> | Layer1 | 84.83 |  74.88 |
> | Layer2 | 76.25 | 79.37 |
> | Layer3 | 63.87 |   86.46 |
> | Layer4 (ReAct) | **31.43** |   **92.95** |
> | No ReAct | 58.41 |   86.17   |
>
>
> **2. Positive skewness of OOD data**
> The assumption of positive skewness is motivated by our observation on the real OOD data. For example, Figure 1(b) shows the OOD activation mean per unit, with a few units having abnormally high values, i.e. positive skewness. This observation is surprisingly consistent across datasets and model architectures. Although we agree that a more in-depth understanding of the fundamental cause of positive skewness is important, for this work, we chose to rely on this empirically verifiable assumption and instead focus on analyzing our method, ReAct. As the reviewer suggested, we have added a discussion on this limitation.
>
> **3. Distributional trend under Gaussian noise**
> Great suggestion! We had indeed verified the activation pattern under Gaussian noise, which appears to exhibit a similar distributional trend (with positive skewness and chaoticness). We have included the plot in the final version for completeness.
>
> **4. Open-source code**
> We are very much in support of open-sourcing code for the community to reproduce and build on our work. We have gone through a thorough codebase cleanup and code documentation and will attach the Github link in our final version. Thank you again for pointing that out!

---

### Official Review · Reviewer_phHd · 2021-07-20

**Rating:** 7
**Confidence:** 5

**Summary:**

This proposes ReAct to reduce model overconfidence on OOD data.
This paper observes the mean activation of OOD data in the penultimate layer has larger variations across units and is biased towards having sharp positive values, which produces overconfident predictions on OOD data.
ReAct rectifies the activations at an upper limit for OOD data and preserves the activation for ID data.
The results show that ReAct makes output distributions between ID and OOD data more separable.

**Limitations And Societal Impact:**

-- ReAct performs well on the classification networks. Can ReAct help OOD detection in other computer vision tasks, such as detection and segmentation networks?

-- As  OOD data can trigger unit activation patterns that are significantly different from ID data. Why not detection OOD data from such separable unit activation patterns directly?


**Main Review:**

The overall methods is simple and effective to detect OOD data.
Extensive expermental and theoretical analysis are given.

**Time Spent Reviewing:**

4

---

> ### Author Response · Authors · 2021-08-10
> **Thank you for the encouraging feedback**
>
> We are encouraged that the reviewer found our method simple, effective, and supported by extensive experiments and analysis. We thank the reviewer for the helpful comments and suggestions, which we address below:
>
> **1. Effect of ReAct in other computer vision tasks**
> Excellent suggestion! As the reviewer recognizes, our method is a simple plug-and-play solution for common neural networks, and hence has the potential for tasks beyond image classification. We focused on image classification as it’s directly motivated by the prevalence of overconfidence issues, as established since [Nguyen et al. 2015]. Moreover, OOD detection literature has also been commonly evaluated on the image classification task, hence we thought it’d be a natural setting to explore. We believe it’s much easier for the community to build on our work if we focus on an established task, understand it well and thoroughly before evaluating it on other tasks. With that being said, we certainly agree with the importance of extending our work beyond image classification. We believe it can be a worthy follow-up work where one can delve much deeper into the object detection/segmentation, carefully construct the evaluations and thoroughly compare the performance. We have added a paragraph acknowledging this limitation and potential future direction. Thank you again for the suggestion!
>
>
> **2. Why not detecting OOD data using the separable unit directly**
> This is a very interesting suggestion! Since each OOD test dataset may trigger a different subset of units with unusually high activation, it is difficult to anticipate such a subset in advance. This is in fact the main challenge in OOD detection without too much prior knowledge on the OOD test datasets. Our method is OOD-agnostic without relying on the specific subset of units that can be triggered by OOD data. Therefore, it is more general and also easy to use.
>
> These discussion points have been added to our draft. Thank you again for pointing them out!

---

### Author Response · Authors · 2021-08-10
**Summary of author response - thank you for the valuable and constructive feedback**

We thank all the reviewers* for constructive and valuable feedback. We are glad that reviewers find our method **simple**, **novel**, and **effective** (R1, R2, R3, R4), **well-motivated** (R4), **well-written** with **clear arguments** (R2, R4), and **easy to implement and use** (R2). The reviewers also agreed that the **experiments & ablations are extensive** (R1, R4), with **compelling** and **strong results** (R1, R2, R4). Moreover, we are more than encouraged that the reviewer found the analysis **insightful and useful for research beyond OOD detection** (R4).

We have addressed the reviewers’ comments and concerns in individual responses to each reviewer. The reviews allowed us to strengthen our manuscript and the changes made are summarized below:

+ [R1, R2, R3] Added discussion on limitations and potential impact as suggested by the reviewers.
+ [R2] Added numerical results on the effect of different layers
+ [R2] Added a new plot of unit activation patterns under Gaussian noise, as well as the link to open-source code.
+ [R3] Added performance of individual OOD test datasets for CIFAR in the appendix.
+ [R3] Added details of Gaussian noise.
+ [R3] Added analysis and discussion training-time regularization approaches; fixed misplacement of citation [14].
+ [R4] Revised the abstract and introduction to highlight that ReAct generalizes to different network architectures and works with different OOD detection methods.
+ [R4] Added clarification for Fig 2, and for why ReAct improves OOD scoring functions.

(\* As abbreviations, we refer to reviewers **phHd** as R1, **68cE** as R2, **hJUX** as R3, and **QB7D** as R4 respectively.)

---

### Public Comment · ~Thomas_G_Dietterich1 · 2021-11-30
**Questions about this paper**

We read this paper in our group at Oregon State, and we have several points of confusion. Perhaps you can address them in the final version (or at the poster session).
1. What is the definition of FPR95? In the MOS paper [18] it is the false positive rate of OOD examples when the true positive rate for ID examples is 95%. Do you set the classification threshold at the 95th percentile of the ID scores? In most OOD research papers, "positive" means OOD, so we found this confusing.
2. We needed help understanding Figure 1. In (a) and (c), you say these are uncertainty scores. Is that the max_y P(y|x), so that small values are more uncertain? Presumably the decision threshold (the left edge of the hashed region) is the FPR95 threshold, right? This would actually be the 5th quantile of the uncertainty score. In (b), and in other figures (e.g., Fig 5) in the paper, we can see a horizontal line in the top plot. Is that the zero axis? If so, how can we have variances that go negative? Are the vertical axes the same? If so, why isn't there a similar horizontal line in the bottom plot?
3. We compared the results in Table 1 to the results in Table 1 of [18]. Why are your results better? [18] uses a better supervised network, and generally, OOD performance improves when the supervised network improves.
4. In 4.1, you say "For a fair comparison, all methods use the pre-trained networks post hoc." But is that a fair comparison? Many OOD methods (such as G-ODIN or DeepHybrid), try to learn a better representation. What is the state of the art in this area?
5. How did you tune the ODIN parameters (temperature and perturbation step size)?
6. We didn't understand why the "batchnorm oracle" statistics improve separability. Figure 4 suggests that the batchnorm oracle exhibits less of the large spiking behavior that your method corrects. Won't the activation statistics for ID and OOD be much more similar with oracle batchnorm parameters? It would seem that because WeightNorm and GroupNorm exhibit the same OOD behavior, the BatchNorm explanation is wrong. Clearly we are confused :-)
7. Did you try using a different c for each unit? It would be easy to do, and it might improve performance. It looks to our eyes as if some of the OOD spikes correspond to smaller ID spikes, so having a separate c could help capture that signal rather than truncating them both.
8. We work primarily on open category detection. There, we (and others, e.g., Vaze, et al. arXiv:2110.06207; Tack, et al. 2007.08176) find that activations for novel categories are smaller than for known categories. This seems to be the opposite of what you found. Any ideas about what is going on?

---

> ### Public Comment · Authors · 2021-12-01
> **Response**
>
>
> Thank you for your interest in our work. Please see our response to your insightful questions below:
>
> 1. Yes, the definition of FPR95 is the same as in [18]; see Section 2 in our paper. The threshold is chosen so that 95% of the ID scores are above it. This definition aligns with the convention that larger scores (say, confidence) indicate more ID-ness and vice versa.
>
> 2.
>    - The uncertainty score that we referred to in the caption of Figure 1 is the negative energy score [33], that is, $\text{logsumexp}(f^c(x))$ where $f^c(x)$ is the logit output for class c. We will clarify this in a revision. The uncertainty score is consistent with our main results in Table 1.
>    - Yes, the decision threshold (the left edge of the hashed region) is the FPR95 threshold which is set when the true positive rate for ID examples is 95%. So it is the 5th percentile of ID's score (blue distribution).
>    - Thanks for pointing out the plotting glitch. The thin horizontal line should align with the x-axis (zero axis) for the OOD plot.
>    - The shaded area shows activation (mean-std, mean+std), which can be negative since the distribution of activation values is asymmetric with the minimum 0. Admittedly, a better way to visualize the distribution of activations might be top-/bottom-25th percentile, which does not become negative.
>
> 3. The ResNet model used in [18] is ResNet-101 with image resolution 480 x 480. What we reported in Table 1 is ResNet-50 with resolution 224 x 224. The fact that ReAct outperforms MOS (despite the smaller model capacity in ReAct) demonstrates the methodological advantage. MOS requires retraining using a different loss function, whereas ReACT mitigates the overly high activations in the feature space as we do.
>
> 4. By "post hoc" and "fair comparison", we meant to highlight the fact that all methods derive OOD scores from a pre-trained model using softmax cross-entropy loss---one of the most commonly used training objectives. We believe that post hoc methods have greater applicability because:
>    - It does not cause performance on the ID task to deteriorate significantly.
>    - Re-training under a different loss function can be sometimes disruptive to the ML pipeline in industry applications, so it is more realistic to perform OOD detection using pre-trained networks.
>
>     We additionally evaluated using G-ODIN, where the average FPR95 is 66.07 on ResNet (same evaluation setting as Table 1). In contrast, ReACT's FPR95 is 31.43 (average across four datasets).
>
>     On a separate line where we can modify the training objective, recent development in representation learning and contrastive learning [A, B, C] shows encouraging performance.
>
>     [A] Jim Winkens, Rudy Bunel, Abhijit Guha Roy, Robert Stanforth, Vivek Natarajan, Joseph R Ledsam, Patricia Mac Williams, Pushmeet Kohli, Alan Karthikesalingam, Si-mon Kohl, et al. Contrastive training for improved out-of-distribution detection. arXiv preprint arXiv: 2007.05566,2020
>
>     [B] Jihoon Tack, Sangwoo Mo, Jongheon Jeong, and Jinwoo Shin. Csi: Novelty detection via contrastive learning on distributionally shifted instances. In Advances in Neural Information Processing Systems, 2020
>
>     [C] Vikash Sehwag, Mung Chiang, and Prateek Mittal. Ssd: A unified framework for self-supervised outlier detection. In International Conference on Learning Representations, 2021.
>
>
>  5. In all experiments, we set the temperature scaling parameter T= 1000. For ImageNet, we found the input perturbation does not further improve the OOD detection performance and hence we set ε = 0.
>
>  6.
>    - Excellent question raised there. We'd like to highlight that similarity from activation statistics can be insufficient to determine separability. Consider a toy example, ID = {(-1, 1), (1, -1)} and OOD = {(1,1), (-1,-1)} have the same activation statistics but are fully separable.
>
>    - We would like to clarify that the BatchNorm explanation is not general to all networks, but only ones trained using BatchNorm. The fact that ReAct applies to models trained using WeightNorm and GroupNorm does not contradict this explanation, since ReAct is not attempting to correct the BatchNorm statistics, but rather tackle the spiky activation problem caused by it (and other normalization schemes).
>
> 7. Great suggestions! Unfortunately, it is hard to anticipate which spikes for ID or OOD will be higher or smaller since they are test datasets and different test sets could produce different spikes.
>
>  8. Our observation concurs with what you said, that OOD activations are smaller than ID activations on average. For instance, Figure 1 shows the OOD activation mean is lower than the ID activation mean.
>
> If you have any further questions, we'd be happy to address them at the poster session!

---

### Decision · Program_Chairs · 2021-09-27

**Decision:**

Accept (Poster)

**Comment:**

This paper observes that OOD samples tend to have overconfident activations and it proposes a simple activation clipping mechanism to reduce model overconfidence on OOD data. Although the proposed method is simple and straightforward, the reviewers found the theoretical analysis and insightful discussion valuable to the research field. Given the strong results and clear presentation of the ideas, I recommend acceptance.